# Dual Optical Path Based Adaptive Compressive Sensing Imaging System

**DOI:** 10.3390/s21186200

**Published:** 2021-09-16

**Authors:** Hongliang Li, Ke Lu, Jian Xue, Feng Dai, Yongdong Zhang

**Affiliations:** 1School of Engineering Science, University of Chinese Academy of Sciences, Beijing 100049, China; luk@ucas.ac.cn (K.L.); xuejian@ucas.ac.cn (J.X.); 2Institute of Computing Technology, CAS, Beijing 100190, China; fdai@ict.ac.cn; 3School of Information Science and Technology, University of Science and Technology of China, Hefei 230027, China; zhyd73@ustc.edu.cn

**Keywords:** compressive sensing, imaging system, neural networks

## Abstract

Compressive Sensing (CS) has proved to be an effective theory in the field of image acquisition. However, in order to distinguish the difference between the measurement matrices, the CS imaging system needs to have a higher signal sampling accuracy. At the same time, affected by the noise of the light path and the circuit, the measurements finally obtained are noisy, which directly affects the imaging quality. We propose a dual-optical imaging system that uses the bidirectional reflection characteristics of digital micromirror devices (DMD) to simultaneously acquire CS measurements and images under the same viewing angle. Since deep neural networks have powerful modeling capabilities, we trained the filter network and the reconstruction network separately. The filter network is used to filter the noise in the measurements, and the reconstruction network is used to reconstruct the CS image. Experiments have proved that the method we proposed can filter the noise in the sampling process of the CS system, and can significantly improve the quality of image reconstruction under a variety of algorithms.

## 1. Introduction

With the development of mobile Internet and imaging technology, a large number of images are generated in daily life. As the resolution of the images keeps increasing, the density of the photoelectric sensor will be higher and higher. However, the signal acquisition needs to meet the classic Shannon–Nyquist sampling theorem, which means that, if you want to restore the sampled signal without distortion, the sampling frequency needs to be greater than twice the highest frequency in the signal [1]. For this reason, high-resolution image sensors will generate a large amount of data when taking images. Normally, we will compress these image data with some compression formats, such as the well-known JPEG and PNG. In this process, a large amount of redundant data will be removed, and which will also consume a lot of computing resources. For this reason, we urgently need a method that can directly sample the compressed data. Compressive Sensing [2] is just such a theory that can sample a small amount of compressed data to reconstruct high-quality images.

Compared with the traditional sampling compression theory, the biggest advantage of CS is that the sampling and compression processes are carried out at the same time, and the compressed data can be collected directly. However, the theory assumes that the sampled signal needs to meet a certain sparsity. The proposal of this theory enables signal acquisition to break through the Shannon–Nyquist sampling theorem under certain conditions, and can restore the original signal by sampling a small amount of data, which makes the theory have a wide range of applications in practice, such as medical imaging, remote sensing imaging, and sensor networks.

In the practical application of compressive sensing, a spatial light modulator is used to obtain the measurements of the image by displaying the measurement matrix. However, as the difference between the rows of the measurement matrix is very small, the brightness of the light after being modulated by the spatial light modulator is actually similar, which requires a high-sensitivity photoelectric sensor to sample. However, this sensor will fluctuate greatly in response to small changes of ambient light. At the same time, we usually need to perform a high gain of the electric signal in the imaging system. For example, in our hardware device, mentioned later in this paper, the gain is up to $10^{8}$ times. Such a large gain will cause the noise signal to be amplified many times, and even drown the strength of the original signal. In addition, in order to improve the collection speed of the equipment, it is usually necessary to switch the measurement matrix image at a high rate with the spatial light modulator. This requires a high synchronization between the photodiode and the spatial light modulator. The small differences will cause the photodiode to fail to collect the right light signal. In summary, the imaging quality of current CS imaging systems is very poor.

The imaging quality of several imaging systems is shown in Figure 1. Among them, Figure 1a shows the image quality of the single-pixel camera (SPC) [3]. Although there is only a simple letter, and the image has only black and white colors, the edges of the image are blurred. Figure 1b is the imaging quality of Reconnect, and Figure 1c is the result of our imaging system with a 25% measurement rate. We can see that the image quality is very poor. This image quality directly affects the application of CS imaging systems. For this reason, we urgently need a CS imaging system that can adapt to its own noise.

For this reason, we propose an adaptive CS imaging system based on a dual optical path. The main contributions are summarized as follows:(1)Aiming at the problem that deep learning data cannot be labeled in the actual CS sampling, a dual light path acquisition and labeling method is proposed, and this method is used to obtain the data required for network training;(2)A filtering method for a CS imaging system based on deep learning is proposed. At the same time, a reconstruction network is constructed on the basis of this filtering method to better reconstruct CS measurement data;(3)A CS imaging system is established to adaptively filter the noise created by the hardware, which can significantly improve the imaging quality so that the CS imaging device can be better applied to image acquisition.

Experiments show that this filtering method not only has good reconstruction results based on deep learning reconstruction methods, but can also improve the reconstruction quality of other traditional algorithms.

The organization of this paper is as follows: Section 2 briefly introduces the basic concepts of CS and existing imaging systems. In Section 3, the dual optical path acquisition system and the reconstruction algorithm are proposed in detail. The experimental results are shown and analyzed in Section 4. Finally, the conclusion of this paper is presented in Section 5.

## 2. Basic Concepts and Related Work

The related work will be reviewed from the perspective of the basic principles of CS imaging and some imaging systems.

### 2.1. Compressive Sensing

Compressive Sensing is a theory of signal compression acquisition. Its basic principle is to project sparse signals into a low-dimensional space through a special measurement matrix for acquisition. This model was proposed by Donoho, Candes and Tao in their work [2] in 2006. The basic mathematical model is expressed as follows: Assuming the signal to be compressed is $x\in R^{N}$, then the sampling process can be defined as the following equation:(1)y=Φx,
where $\Phi \in R^{M\times N}$ is called the measurement matrix, which is to project the original signal into a low-dimensional space. $y\in R^{M}$ is the compressed signal, called the measurements. The dimension of measurements is much smaller than the dimension of the original signal, which is M≪N.

The reconstruction of CS is a process of recovering an approximate solution of the signal x through a certain algorithm using the measurements y. As we know, the dimension of y is much smaller than the dimension of x. Therefore, the solution from y to x is an NP-hard problem [4]. For practical problems, it is usually necessary to introduce other assumptions to solve this problem.

Traditional CS imaging reconstruction algorithms usually consider the structural characteristics of the image, such as the total variation (TV) model. The minimum total variation model assumes that the image has sparseness in vertical and horizontal gradients, so this method is widely used in CS image reconstruction [5,6]. For the solution of this method, TVAL3 [7] is a recently proposed TV-based CS image reconstruction algorithm. This algorithm not only has a fast convergence speed, but also supports a variety of measurement matrices. It is a commonly used algorithm that can recover image details. The reconstruction algorithm based on the image structure characteristics also includes the NLR-CS [8] method. This method is based on the similar characteristics of non-local images and reconstructs CS images according to the low-rank characteristics of non-local image blocks. This algorithm is also a reconstruction algorithm based on image structure characteristics. There are also some algorithms based on image filters, among which D-AMP, proposed by [9], is more widely used.

In recent years, some scholars have used neural networks to solve the problem of CS reconstruction. DNN was first introduced into CS recovery by Mousavi in the work on the stacked denoising autoencoder (SDA) [10]. The work proposed an autoencoder based on multi-layer perceptions (MLPs) to solve the CS recovery problem. Inspired by the work on super-resolution (SRCNN) [11], Kulkarni et al. proposed a CNN-based neural network called ReconNet in their work [1]. This work first introduced convolution neural networks (CNN) to CS and the performance is better than that of SDA. However, these methods do not consider the structural characteristics of the image, such as the saliency of the image. Compressed sensing based on blocks can more effectively process natural images [12,13]. These methods divide the image into non-overlapping blocks and perform separate measurements on different image blocks. In [14], fewer measurement rates are allocated to non-salient image blocks but more are allocated to salient blocks. This can effectively improve the efficiency of compressive sensing measurements, thereby improving the reconstruction quality of the saliency area. However, this method will cause obvious blocking artifacts. In [15], a multi-channel deep network was proposed for a full image to remove blocking artifacts. AMP-Net [16] iteratively uses denoising and deblocking methods to eliminate block artifacts in the image. Since the actual light cannot be accurately divided into blocks, these methods cannot be applied to actual compressive sensing image acquisition.

There are also several works based on the recent, popular Generative Adversarial Networks (GAN). The work [17] added a discriminator to the network of ReconNet. Dave et al. implemented the generative model RIDE [18] for pixel prediction to recover the compressive image in [19]. This work reconstructed the image pixel by pixel, which is a new type of reconstruction method. In [17], a fully connected layer was added to the ReconNet architecture to learn the measurements from the scene image. The work in [20] used the method of a joint learning measurement matrix, which greatly improves the reconstruction quality. These methods of learning measurement matrices can obtain matrices that are more suitable for natural images, but the training process for these matrices is complicated and at the same time highly dependent on the training set. In recent years, DNN has also been applied to CS video recovery [21,22].

### 2.2. Imaging System

The basic theory of CS sampling is y=Φx, where x is the light in the scene we need to sample and Φ is the measurement matrix. The key problem is how to perform matrix multiplication on the light and how to obtain the measurements. We know that the spatial light modulator can control the amount of light passing through different pixel positions to achieve modulation of the light, such as the commonly used devices: liquid crystal panels, digital micromirror devices (DMD), and so forth. The modulated light is received by the photoelectric sensor to obtain the measurements.

Using DMD to modulate light, Duarte proposed a CS imaging system: the Single Pixel Camera (SPC) [3]. The light reflected by the objects in the scene is imaged onto the DMD through the convergence of the lens. Then, the DMD modulates the measurement matrix to reflect the modulated light afterwards. Finally, the modulated light is sampled by the photodiode. SPC is the first acquisition system for CS imaging. The system uses one pixel to reconstruct an image with a resolution of 256×256. However, due to the acquisition noise, its imaging quality is poor.

Using the LCD panel as the spatial light modulator, Huang proposed a lensless camera [23]. Using the principle of small aperture imaging, the camera does not require a lens. However, as we know, the lens is used in the traditional camera to converge the light and increase the imaging field of view. As the light transmittance of the liquid crystal is poor, most of the light cannot pass through the liquid crystal, which leads to poor image reconstruction quality.

The above two cameras are based on the traditional camera model, which receives the light reflected by the environment and modulates ir. Some scholars have also proposed an imaging system based on active light. Zhang [24] proposed a system that uses a projector to perform CS imaging. The camera uses a projector to project the measurement matrix into the scene to be sampled. However, this kind of camera collects the light of the entire projection space, which means that other light in the environment will affect the system and increase the noise of the reconstructed image.

At present, CS imaging systems require high sampling accuracy, and generally require high-precision signal amplification and filtering circuits. However, due to the influence of ambient light and circuit noise, the quality of image reconstruction is poor. Therefore, a CS imaging system that can reconstruct images with high quality is needed.

## 3. Adaptive Compressive Sensing Imaging System

In recent years, with the development of deep neural networks, many impressive results have been achieved in the field of image processing. The advantage of neural networks is that they can simulate complex mathematical models, but their disadvantage is that they are data-driven and require a lot of training data. Based on this feature, we propose a system to use neural networks to remove noise in CS imaging hardware and improve imaging quality. Its basic structure is shown in Figure 2. In order to obtain the data needed for neural network training, we designed a data acquisition system based on dual optical paths. Next, we will give a detailed introduction of the basic structure and hardware system of the dual optical path design.

### 3.1. Dual Optical Path Sampling

The purpose of the dual-optical acquisition system is to obtain image data and CS sampling data of the same scene at the same time, so as to calibrate the measurements of CS and optimize the reconstruction results. To achieve this purpose, we use the bidirectional reflection characteristics of DMD. In other words, the DMD micromirror can reflect light at ±12 degrees. Usually in CS imaging hardware, DMD is used to modulate the measurement matrix and the scene light. However, only +12 degrees light is used in this process. In fact, the −12 degrees optical path can also obtain the image information. More importantly, the viewing angle of the scene observed by the ±12 degrees optical path is the same, which provides a guarantee for us to design a dual optical path acquisition and calibration system.

As is shown in Figure 3, the light in the scene is converged by the viewfinder lens and is imaged on DMD. At this time, the image is divided into discrete image pixels by micromirrors on the DMD, where each mirror represents a pixel. The forward-reflected light is converged on the photoelectric sensor through the CS Sampling optical path lens. At this time, if each row of the measurement matrix is displayed one by one on the DMD, we can obtain the measurements y of CS. The reverse light is imaged on the CMOS image sensor through the CMOS light path lens. At this time, if the DMD displays a white image, then the CMOS will capture all the light in the scene, which is the scene image x. In this way, we will obtain a set of label data <x,y>, where y is a set of measurements with the optical path and circuit noise in the CS acquisition hardware, and x is the target image data. At this point, we have obtained the calibration data required for the training of the CS imaging system.

As shown in Figure 4, in order to avoid the influence of the circuit on the optical path, the hardware of the sampling system is divided into two parts: the optical sampling part and the circuit control part.These two parts are shielded from light. In the optical sampling part, the main lens is used to control the aperture, focal length and focus, and so forth, for the convenient sampling of different scenes. DMD uses the D4100 produced by Texas Instruments (TI) with a pixel resolution of 1920×1080. Among them, the CMOS cannot completely align the image on the DMD, and we also only use some of the pixels in the center of the DMD, so the image obtained by the CMOS needs to be calibrated before it can be used. The circuit control part mainly includes the controller of each sampling part and a signal amplifier, among which the amplification factor of the signal amplifier reaches $10^{8}$ times.

### 3.2. Recovery Network

To adaptively filter the device noise, we need to use the characteristics of neural networks to train the noise model of the imaging device. After this, the filtered measurements are obtained, which are then reconstructed to improve the imaging quality of the device. For this reason, the design of the neural network is divided into a filter network and a reconstruction network.

#### 3.2.1. Filter Network

Considering that the poor imaging quality of the CS imaging system is mainly because the sampled measurements contain a lot of noise, we first build a CNN network to denoise the measurements. As shown in Figure 5, a 4-layer denoising network is built based on convolution. This process can be expressed as $\hat{y}=C\left(y,W^{c}\right)$, where $\hat{y}$ is the filtered measurements, and $W^{c}$ is the weights of the convolution layer. The basic structure of the network is shown in Figure 5, where each convolution layer is followed by batch normalization and the ReLU activation function. The first convolution layer takes a kernel size of 11 and outputs 32 feature maps. The second takes a kernel size of 1 and outputs 64 feature maps. The third takes a kernel size of 7 and outputs 32 feature maps. The last layer takes a kernel size of 3 and generates 1×M measurements. To obtain $W^{c}$, MSE is employed as the loss function.

#### 3.2.2. Reconstruction Network

The measurements obtained through the filter network can already be used in the reconstruction in traditional CS algorithms, but in order to obtain better reconstruction quality, we ran a reconstruction network after the filter network. The network is composed of two fully connected layers. Assuming that the approximate value of the image output by the network is $\hat{x}$, then $\hat{x}=L\left(y,W^{l}\right)$, where $W^{l}$ is the weights of the fully connected layers, which are trained by another MSE loss function.

#### 3.2.3. Training

The training of the neural network of the aforementioned imaging system is divided into three parts; the first part is the training of the filter network, the second part is the training of the reconstruction network, and the third part is the joint training. We set the sample image size to 128×128; a total of 44 images were sampled on the device, of which eight images were used as the test set.

##### Filter Network Training

Since the measurements are a 1×16,384 signal, we intercept the measurements with a length of *M*. The specific value of *M* is related to the sampling rate. The lengths of 1%, 4%, 10%, and 25% are 164, 655, 1638, 4096, respectively. After setting the step size to 20, we obtained about 27,000 sets of training data.

##### Reconstruction Network Training

The training of this network is similar to the general training of a CS reconstruction network based on deep learning, such as ReconNet. Use the existing image dataset, we simulate the measurements and then train the network. Here we choose about 14,000 sets of training data.

##### Joint Training

Joint training refers to the combination of the above two networks, the initial loading of the two parts of the network weights that have been pre-trained, reducing the learning rate of the network, and training on 36 datasets sampled by our device. All network training is implemented on pytorch.

The proposed network is implemented on the public Pytorch framework. The weights of the networks are initialized by normal Gaussian distribution with the stand variance of 0.01. The optimizer algorithm for all training is Adam. We run the training model on a PC with 2.1 GHz CPU, 128 GB RAM and a NVIDIA GeForce RTX 2080Ti GPU. The learning rate is initialized with $1\times 10^{-4}$ and decreased by 50% at 20 epochs. The whole training procedure takes 100 epochs for each network. The training time for each method takes about 8 h.

## 4. Experimental Results

### 4.1. Comparison with Existing Methods

In order to verify the imaging effect of our reconstruction network, we compare the reconstruction results of different algorithms at the same measurement rate (MR) and the same measurements sampled by our hardware device. The algorithms include TVAL3 [7], NLR-CS [8] and D-AMP [9]. The PSNR and SSIM results of eight images are shown in Table 1. The reconstruction images of a 25% measurement rate can be found in Figure 6. The quantization of all images uses 8 bits, including the calculation of reconstructed images and PSNR. As the sampled measurements are noisy, some algorithms cannot even reconstruct images, such as partial examples of TVAL3 and NLR-CS. It can be seen that our algorithm has the best reconstruction quality.

Compared with the existing methods, our method has the following advantages:It can be seen from the average value in Table 1 that our method achieves the best reconstruction performance. Our method achieves the best PSNR and SSIM under all the four measurement rates; especially at high sampling rates, the PSNR and SSIM are much higher;Observing the results, we can find that the result of the D-AMP algorithm is the second best, and it is obviously better than the other two algorithms. The main reason is that the D-AMP algorithm is an iterative algorithm based on a filter, and the filter has a certain effect on the noise, but the effect is not very good;The results of TVAL3, NLR-CS and D-AMP at a high measurement rate of 0.25 are not better than those at a low measurement rate of 0.10. The main reason is that the device samples more data at high measurement rates and the acquisition time is long, which will introduce more cumulative errors.

Under the same environment, the average reconstruction time of TVAL3, NLR-CS and D-AMP for each image is 0.23, 30.11, and 8.25 s, respectively. Since the neural network can be accelerated by the GPU, the average reconstruction time of the reconstruction network is 0.02 s. When only the CPU is considered for reconstruction, the average obtained reconstruction time is 0.11 s.

From Figure 6, we can see that the unfiltered data reconstruction result will have some vertical noise in the fixed area of the image, which is present in almost all images. Compared with other unfiltered algorithms, the reconstruction results of D-AMP are a little better, but there is a serious phenomenon of denoising and smearing, which leads to blurry images. Our method filters out noise while preserving details.

### 4.2. Evaluation of Filter Network

To evaluate the performance of the filter network, we first pass the measurements through our filter network, and then reconstruct it with the TVAL3, NLR-CS and D-AMP algorithms again. The results obtained are shown in Table 2. The reconstruction result is the average PSNR and SSIM of eight test images. Comparing with the previous results, we can find the following characteristics:The quality of the reconstruction results of TVAL3, NLR-CS and D-AMP has been significantly improved. The increase in PSNR of some algorithms is about 7 dB;The higher the measurement rate, the greater the improvement in image quality. The main reason is that the measurement rate is high, the device acquisition time is long, and the cumulative error introduced will be large, so the effect achieved by the filter network is better;In fact, the reconstruction result of 0.01 is already very poor; the image basically cannot show the content of the original image. The SSIM of the image is also very low and the PSNR of D-AMP after filtering is reduced, which is meaningless.

At the same time, we compare the difference between the sampled signal, the filtered signal and the original signal. It is obvious that a low-frequency noise signal is added onto the sampled signal, which actually includes some high-frequency signal noise. After the process of our filter network, the signal is basically the same as the original signal, so the reconstruction quality is significantly improved. By comparison, the error between the filtered signal and the original signal is much smaller, indicating that the filter network has a significant effect on the noise of the CS imaging device.

The advantage of the compressive sensing imaging system proposed in this work is the use of dual optical paths for sampling. The system can not only provide the measurements of compressive sensing imaging, but can also calibrate the measurements so that no matter which reconstruction method is used, a better reconstruction quality can be obtained. Experiments have proved that the compressive sensing sampling method based on dual optical paths can achieve better image quality in a variety of reconstruction methods. As shown in Figure 7a, there is obviously a low-frequency noise in the sampled signal. The signal after filtering obviously removes the low-frequency noise signal and is very similar to the original signal. As shown in Figure 7b, the mean square error of the filtered signal is greatly reduced for each image.

## 5. Conclusions and Future Work

Because CS imaging systems have high sensitivity to ambient light and circuit noise, even if we add a variety of hardware filtering methods, their imaging quality is still very poor. To this end, we propose a dual optical path CS imaging system. The design of the dual optical path is based on the bidirectional reflection characteristics of DMD, so that one optical signal is used for CS measurements sampling, and the other optical signal is used for CMOS images acquisition. In this way, we labeled the measurements with the images in the same viewing angle of the scene, and this labeled data are required for neural network training. We designed and implemented a dual optical path imaging system, and also proposed a filter and reconstruction network. After training, this reconstruction network can obtain better imaging results. Through the filter network, the noise in the measurements is suppressed, and the reconstruction quality is significantly improved even for the traditional CS reconstruction algorithms. Experiments have verified that this dual optical path based CS imaging system can effectively improve the imaging quality and suppress the noise caused by the optical path and the circuit on the measurements. The filter network can adaptively filter the noise created by the hardware, so that the CS imaging device can be better applied to image acquisition.

Although the method we proposed can improve the imaging quality of a CS imaging system, the system will take a large amount of image data during its work. Therefore, future work will focus on another technique that is required to use these newly generated image data for training and to continuously update the filter network online.

## Figures and Tables

**Figure 1 sensors-21-06200-f001:**
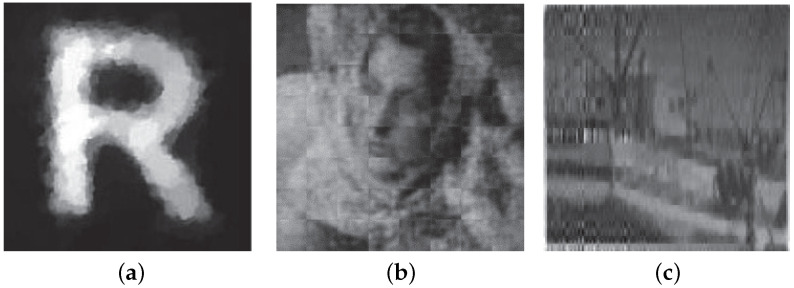
Imaging quality of several systems. (**a**) Single pixel camera; (**b**) ReconNet hardware imaging; (**c**) Our hardware system imaging.

**Figure 2 sensors-21-06200-f002:**
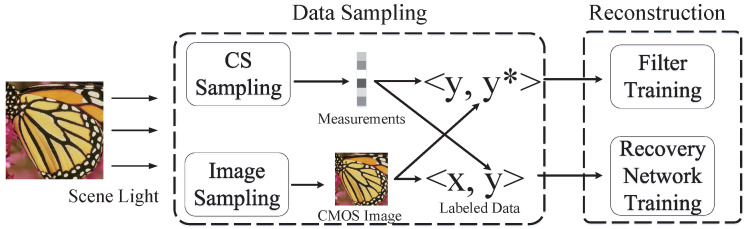
Architecture of dual optical path based adaptive CS imaging system.

**Figure 3 sensors-21-06200-f003:**
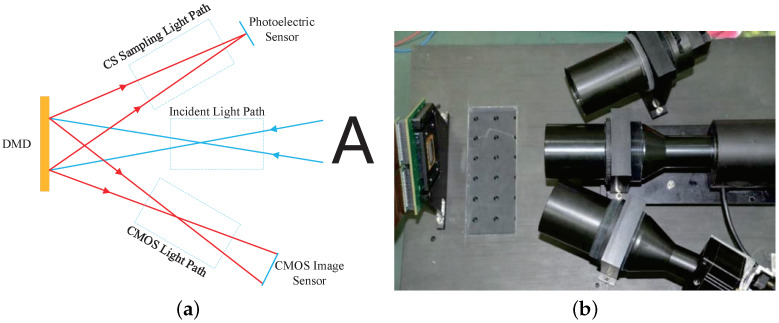
Dual optical path design. (**a**) Optical path design; (**b**) Hardware design.

**Figure 4 sensors-21-06200-f004:**
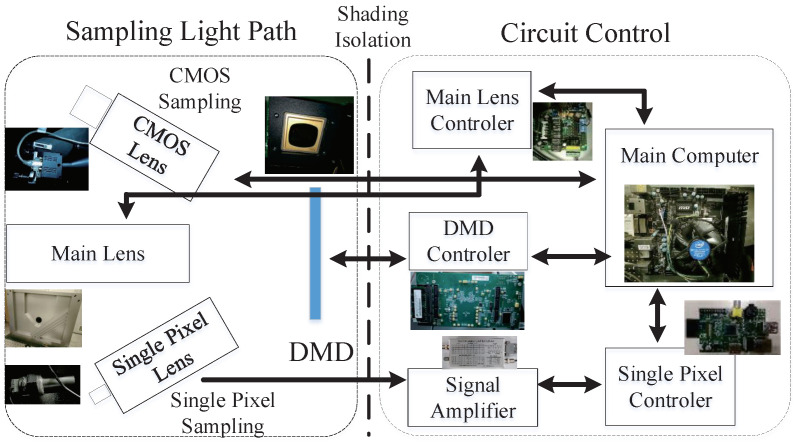
Sampling system architecture.

**Figure 5 sensors-21-06200-f005:**
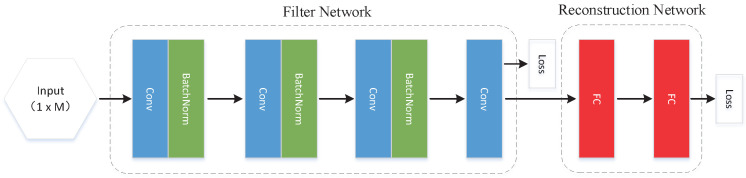
Neural networks of the imaging system.

**Figure 6 sensors-21-06200-f006:**
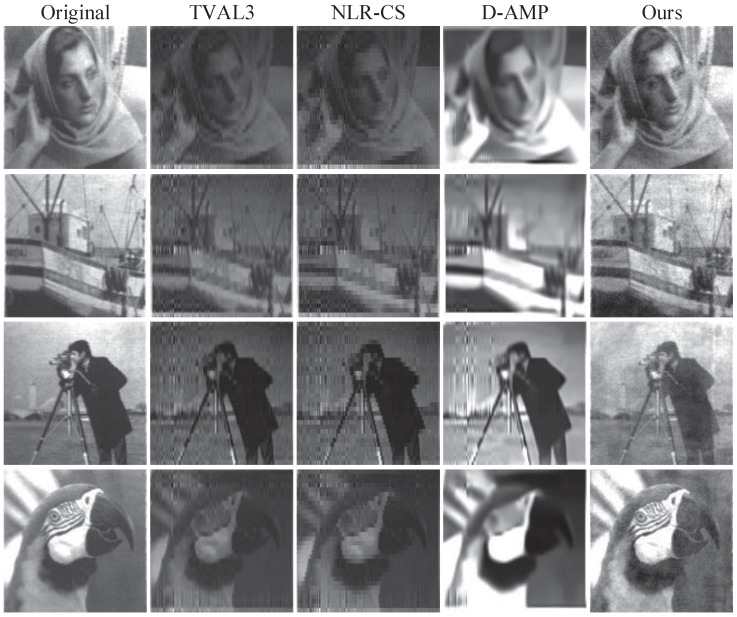
Reconstruction results of the images.

**Figure 7 sensors-21-06200-f007:**
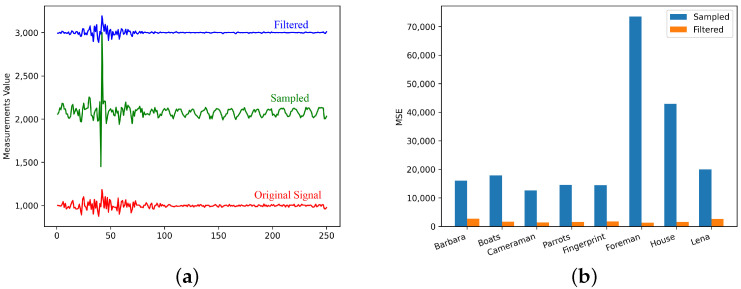
The effect of the filter network on the signal. (**a**) Raw data of sampled signal and filtered signal; (**b**) The mean square error of the sampled signal and the filtered signal.

**Table 1 sensors-21-06200-t001:** The reconstruction results of different algorithms at different MR. The inputs of the algorithms are the data directly sampled by hardware. The best results are in bold. Our algorithm can achieve the best results at each MR.

Image	Algorithm	MR = 0.25		MR = 0.10		MR = 0.04		MR = 0.01	
		PSNR	SSIM	PSNR	SSIM	PSNR	SSIM	PSNR	SSIM
Barbara	TVAL3	9.84	0.43	10.23	0.42	11.66	0.03	12.51	0.23
	NLR-CS	9.79	0.42	10.64	0.39	9.99	0.33	10.96	0.28
	D-AMP	12.80	0.58	12.18	0.49	11.88	0.40	**14.17**	0.32
	Ours	**22.65**	**0.70**	**22.61**	**0.77**	**20.83**	**0.68**	14.01	**0.38**
Boats	TVAL3	9.95	0.44	8.32	0.38	7.45	0.30	6.47	0.20
	NLR-CS	9.57	0.41	7.64	0.32	7.29	0.28	6.05	0.17
	D-AMP	11.98	0.50	11.42	0.41	10.44	0.37	12.66	0.29
	Ours	**20.76**	**0.64**	**19.97**	**0.71**	**17.75**	**0.63**	**15.92**	**0.43**
Cameraman	TVAL3	9.68	0.46	9.85	0.48	11.06	0.44	10.01	0.30
	NLR-CS	9.71	0.43	10.20	0.45	11.12	0.40	10.02	0.30
	D-AMP	12.86	**0.55**	12.49	**0.55**	12.50	**0.46**	11.03	**0.32**
	Ours	**18.80**	0.41	**18.10**	0.50	**17.34**	0.30	**14.39**	0.18
Parrots	TVAL3	10.38	0.0.53	9.53	0.41	9.85	0.41	9.60	0.23
	NLR-CS	10.13	0.50	9.38	0.36	9.35	0.35	9.46	0.20
	D-AMP	11.77	0.54	13.32	0.58	11.91	0.50	10.57	0.27
	Ours	**23.27**	**0.70**	**22.86**	**0.80**	**18.93**	**0.61**	**15.92**	**0.52**
Fingerprint	TVAL3	10.54	0.21	12.57	0.21	10.11	0.11	6.60	0.10
	NLR-CS	10.34	0.20	12.20	0.21	8.08	0.15	6.77	0.10
	D-AMP	12.76	0.23	12.46	0.22	10.63	0.16	**15.33**	0.17
	Ours	**17.79**	**0.30**	**18.14**	**0.34**	**17.28**	**0.29**	14.96	**0.20**
Foreman	TVAL3	3.44	0.06	12.11	0.47	3.46	0.06	4.36	0.19
	NLR-CS	3.45	0.06	5.25	0.29	4.86	0.24	4.32	0.17
	D-AMP	11.40	0.56	13.44	0.59	13.03	0.52	12.40	**0.47**
	Ours	**20.99**	**0.77**	**17.93**	**0.77**	**20.93**	**0.77**	**14.58**	0.36
House	TVAL3	4.23	0.09	12.43	0.43	4.45	0.12	5.45	0.21
	NLR-CS	4.30	0.10	5.86	0.28	5.15	0.21	5.42	0.19
	D-AMP	10.99	0.54	11.50	0.46	11.53	0.49	12.94	**0.44**
	Ours	**24.42**	**0.72**	**17.57**	**0.82**	**17.59**	**0.70**	**16.87**	0.42
Lena	TVAL3	11.81	0.48	14.09	0.25	11.99	0.25	11.79	0.28
	NLR-CS	11.38	0.46	10.96	0.40	10.92	0.37	11.73	0.26
	D-AMP	12.32	0.55	12.01	0.45	11.56	0.42	10.04	0.21
	Ours	**22.14**	**0.76**	**22.18**	**0.87**	**20.51**	**0.76**	**14.52**	**0.28**
**Mean**	TVAL3	8.73	0.34	11.14	0.38	8.75	0.22	8.34	0.22
	NLR-CS	8.58	0.32	9.02	0.34	8.35	0.29	8.09	0.21
	D-AMP	12.11	0.51	12.35	0.47	11.69	0.42	12.39	0.31
	Ours	**21.35**	**0.63**	**19.92**	**0.70**	**18.89**	**0.59**	**15.14**	**0.34**

**Table 2 sensors-21-06200-t002:** PSNR and SSIM of traditional algorithm after filter.

Image	Algorithm	MR = 0.25		MR = 0.10		MR = 0.04		MR = 0.01	
		PSNR	SSIM	PSNR	SSIM	PSNR	SSIM	PSNR	SSIM
**Mean**	TVAL3	16.29	0.48	15.81	0.37	14.73	0.34	10.82	0.19
	NLR-CS	16.19	0.46	15.66	0.39	14.43	0.32	10.61	0.22
	D-AMP	17.55	0.50	16.86	0.46	15.25	0.32	10.44	0.17
	Ours	**21.35**	**0.63**	**19.92**	**0.70**	**18.05**	**0.50**	**15.14**	**0.34**

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
