# Peer review of "Dual Optical Path Based Adaptive Compressive Sensing Imaging System"

_sensors, 2021, doi:10.3390/s21186200_

Round 1

Reviewer 1 Report

In this paper, a dual optical path compressed sensing imaging system is proposed, which collects photoelectric information and image information separately, which is beneficial to improve the processing speed of compressed sensing. In addition, deep learning based compressed sensing denoising filtering network and reconstruction network are proposed, and MSE loss joint training is adopted to improve the image quality. The designed experiments show the effectiveness and efficiency of the proposed method comparing with state-of-the-art methods. The organization, writing skills and quality of the article need to be further improved, and some modifications are needed before publication.

There are some advices as:

  1. The experiment should discuss the advantages of dual optical path imaging system. In addition, ablation experiments need to be designed to prove the effectiveness of the reconstructed network.
  2. It is recommended to use table or function form to describe the concrete structure of filter network and reconstruction network. The training section of the neural network should detail the implementation details. (e.g., optimizer selection, epochs, learning rate decay, etc.)
  3. The related work and references of the paper need to be further improved, introducing state-of-the-art as much as possible and point out the deficiencies.
  4. For the improvement, the following papers can be considered to enhance the theory of this manuscript.

Zhang, Dengyong; Yin, Ting; Yang, Gaobo; Xia, Ming; Li, Leida; Sun, Xingming. Detecting image seam carving with low scaling ratio using multi-scale spatial and spectral entropies. Journal of visual communication and image representation. 2017

Zhou, Shuren; Ke, Maolin; Luo, Peng. Multi-camera transfer GAN for person re-identification. Journal of visual communication and image representation. 2019

Reviewer 2 Report

This work proposes a dual-optical imaging system, in the context of Compressive Sensing, which uses bidiretional reflection characteristics of DMD. A neural network is considered to filter noisy measurements. Experiments were performed, and the proposed method could filter the noise in the sampling process of the CS, and the quality of the reconstructed image ouperformed other schemes found in the literature.

Authors should review the text, for example, "formatss" in the first paragraph of Section 1.

The firt paragraphs of Section 1 lacks of references.

Authors argue that the CS-based image quality is poor, more specifically regarding to hardware implementations, in the third and fourth paragraphs. However, they could include this part in a smoothy way, adding more context in order to improve the flow of the text.

The title of Section 2 could be changed by "Basic Concepts and Related Work".

Regarding to the Sections 2.1 and 2.2, authors should point out their contributions with respect to the literature discussed.  

Did the authors consider the quantization of images? How many bits were employed?

Since a neural network was employed in the reconstruction process, authors could evaluate the computational complexity in the reconstruction procedure. Do the proposed method has gains both in reconstruction quality and in the computational complexity, when compared against these other methods?

Round 2

Reviewer 1 Report

Thank you for your submission.

Paper is accepted by reviewer.

Reviewer 2 Report

I am satisfied with this new version of the manuscript. However, I just have two minor issues, in order to enrich the final version of the paper:

1- authors could change "8bit" by "8 bits" when they explain the quantization;

2- authors could change the phrase "Only use the CPU for reconstruction, the average reconstruction time is 0.11 seconds" by "When only the CPU is considered for reconstruction, the average obtained reconstruction time is 0.11 seconds".